# Deep Leaning Based Frequency-Aware Single Image Deraining by Extracting Knowledge from Rain and Background

Yuhong He [1], Tao Zeng [2], Ye Xiong [1], Jialu Li [3] and Haoran Wei [4,*]

1   School of Computer and Information Engineering, Jiangxi Normal University, Nanchang 341000, China
2   School of Software, East China Jiaotong University, Nanchang 330013, China
3   G. Brint Ryan College of Business, University of Noth Texas, Denton, TX 76203, USA
4   Department of ECE, University of Texas at Dallas, Richardson, TX 75080, USA
*   Correspondence: haoran.wei@utdallas.edu

**Abstract:** Due to the requirement of video surveillance, machine learning-based single image de­raining has become a research hotspot in recent years. In order to efficiently obtain rain removal images that contain more detailed information, this paper proposed a novel frequency-aware single image deraining network via the separation of rain and background. For the rainy images, most of the background key information belongs to the low-frequency components, while the high-frequency components are mixed by background image details and rain streaks. This paper attempted to decouple background image details from high frequency components under the guidance of the restored low frequency components. Compared with existing approaches, the proposed network has three major contributions. (1) A residual dense network based on Discrete Wavelet Transform (DWT) was proposed to study the rainy image background information. (2) The frequency channel attention module was introduced into the adaptive decoupling of high-frequency image detail signals. (3) A fusion module was introduced that contains the attention mechanism to make full use of the multi receptive fields information using a two-branch structure, using the context information in a large area. The proposed approach was evaluated using several representative datasets. Experimental results shows this proposed approach outperforms other state-of-the-art deraining algorithms.

**Keywords:** deep learning; machine learning; knowledge extraction; single image deraining; frequency aware; discrete wavelet transform; attention mechanism





## 1. Introduction

Rainy environments often lead to a series of visibility degradations. The presence of rain causes strong light fluctuations which blur the background scene and change the content and color of the image. Some rain image samples are shown in Figure 1. Common types of rain include rain streak, rain-mist and raindrop. Rain streak will present as a bright line on an image and tends to concentrate on the high frequency components of images. Rain-mist is usually generated by the accumulation of rain streak, and will contaminate both low frequency and high frequency components of images. Raindrop has different shapes due to a transmission change of the window or lens [1]. All these types of rain cause spatially variant image degradation [2]. Image distortion is often accompanied by the failure of many computer vision tasks, such as video surveillance [3–5] and autonomous driving [6–8]. Therefore, it is very important to study image restoration in rainy weather environments.

In the past decades, various machine learning-based approaches for deraining have been proposed [9–14]. Traditional machine learning-based approaches treat the single image deraining task as a signal separation problem between the rain pattern and the background pattern. These approaches are based on different basic machine learning models, such as frequency domain representation, sparse representation, Gaussian mixture model, etc. In recent years, deep learning-based approaches have learned the mapping between rainy

images and the corresponding clean image pairs for the deraining task [15–17]. Though these deep learning-based approaches can achieve better performance than traditional approaches, there is still room for further improvement.

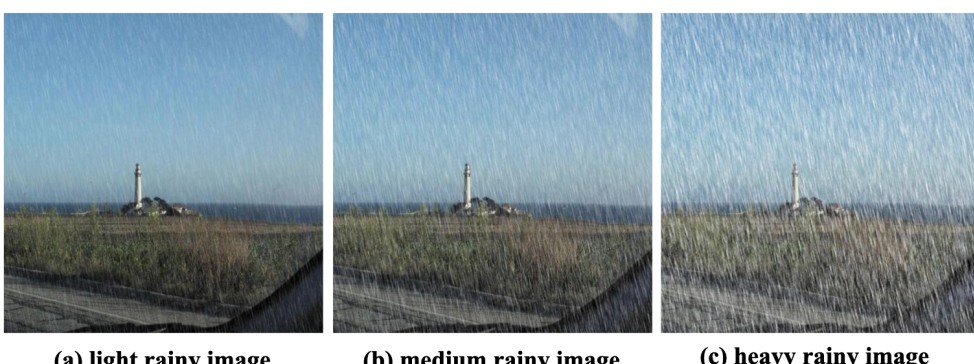

**(a) light rainy image**  **(b) medium rainy image**  **(c) heavy rainy image**

**Figure 1.** Rainy image samples.

First, many previous studies directly learnt the mapping relationship of clear-rain data pairs without considering the inherent characteristics of rain streaks, and the final rain removal images were obtained by simply subtracting the rain streaks from the images with rain. This approach can result in the inadequate removal of rain streaks or the accidental removal of background information coupled to the rain streaks. Second, most previous approaches have a limited receptive field, and it is difficult to obtain contextual information from large regions. Third, many deraining approaches have built complex network structures by stacking modules together to achieve a better outcome of deraining images. This kind of design increases the network parameters and is not efficient. Meanwhile, the stacking of modules causes a poor interpretability of the model, which is inconvenient for further improvements.

To solve these issues, a frequency-aware network and an adaptively selected two-branch structure were proposed to efficiently obtain deraining images. The contributions can be summarized as follows:

(1) A frequency-aware single image deraining network was proposed. This proposed network framework decomposes the single-image rain removal task into two subtasks, with rain-free region background information extraction and rain-obscured region signal recovery, so the network can better preserve background detail.

(2) To solves the problem that the continuous region without rain needs a large perceptual field, the image is converted from the spatial domain to the frequency domain by performing discrete wavelet transformation (DWT). Meanwhile, the proposed attention mechanism allows the network to extract rain-free background information in the low-frequency band, and this provides a reference for the image recovery of regions with rain.

(3) Frequency channel attention was introduced to adaptively enhance or weaken the signals of different frequency bands to better complete the task of image recovery in rain-affected regions.

(4) Experiments conducted on four commonly used publicly deraining datasets shows the efficiency of the proposed approach. Compared to previous popular approaches, the proposed approach achieved a better balance of performance and efficiency.

## 2. Related Works

In the past few years, many excellent rain removal approaches were proposed. This section provides a review on the approaches of single image deraining.

### 2.1. Traditional Methods

Research on the single image deraining task began in 2012, and early approaches mainly used a priori knowledge to represent the features of the background and rain layers.

The development of model-based approaches was driven by the following ideas: image decomposition, sparse coding and prior-based Gaussian mixture models.

Lin [18] proposed an image decomposition method, which is based on a morphological component analysis. This approach first decomposes a single image with rain to extract the high frequency layer of the image, the extracted high frequency layer is further decomposed into two constituents with and without rain using dictionary learning and sparse coding algorithms. Although this approach can recover some of the light rain present in the image, this approach leads to the blurring of some details due to the preprocessing of bilateral filtering. In subsequent work, Luo [19] improved the study of rain sparsity and introduced mutual exclusion in discriminative sparse coding (DSC). Therefore, the background layer can be better separated from the non-linear composite components. Though this approach maintains clear texture details, some rain patterns appear in the output images.

Then Zhu [20] proposed an iterative rain removal approach, which uses the prior information of the rain streak layer to recover the details of the background texture. This approach achieved better results on some synthetic datasets, even with comparable performance to some deep learning-based approaches proposed in the same period. However, the approach cannot achieve satisfactory results when processing real rainy images because the rain streaks in real images move to different directions. Li [21] used Gaussian mixture models (GMMs) to model rain streaks and background layers. This approach can effectively remove small and medium scale rain streaks, but cannot handle large scale rain streaks.

These traditional approaches based on prior knowledge often rely on some specific assumptions, but such assumptions do not always work well for the complex rain situations in a real scenario.

### 2.2. CNN-Based Methods

In 2017, the single image deraining task entered a data-driven era. Due to the powerful feature representation capability of deep learning, deep learning-based approaches have become popular [22–27]. Different kinds of network architectures, such as convolutional neural networks (CNN) [28–30], recurrent neural networks (RNN) [31,32], and generative adversarial networks (GAN) [33,34], have been widely utilized. The embedding of these new models has led to significant performance gains in deraining tasks. However, these approaches rely on the statistical analysis of a large number of rain streaks and background images data.

Yang [35] first proposed a joint rain streak detection and removal network (JORDER). This network can handle many types of rain situations such as heavy rain, overlapping rain streaks, and water accumulation. However, this approach generates under-exposure problems and loss of vertical texture in some background scenes. In the same year, Fu [36,37] attempted to remove rain using a Deep Detail Network (DetailNet). The approach only utilized high frequency detail information as the input. It was demonstrated that learning only rain residuals is beneficial for deraining because it is relatively sparse, which can lead to easier and stable training. However, this approach cannot handle images with heavy rain.

Further studies proposed many new deep learning-based approaches [38–41]. These approaches utilized more recent network structures and introduced new prior information related to rain streaks. These approaches have better results in both evaluation metrics and generated image quality. However, when dealing with some real, unseen images during training, the deraining results are usually not good due to the limitations of supervised learning. Additionally, most of these approaches use some existing network modules in deep learning, then by training the model in an end-to-end manner, inherent prior structure within the rainy images can be ignored.

### 2.3. Semi-Supervised or Unsupervised Methods

In the last two years, to improve the generality and scalability of model architectures, many semi and unsupervised learning methods have been attempted to learn directly from

real rainy images. Wei [42] proposed a semi-supervised learning method using paired synthetic images and unpaired real images for learning. Although this idea is advanced, the algorithm does not perform as expected, especially with real rainy images.

In [43], the unsupervised de-constrained generative adversarial network (UD-GAN) extracts information from two unpaired images by introducing a self-supervised constraint. This approach consists of two collaborative modules. One is used to detect the difference between the real rainy image and the generated rainy image, the other is used to adjust the brightness of the generated image to enhance the visual effect of deraining results. This method can better remove rain from real rainy images, but loses some background detail.

Meanwhile, other works [44,45] combined the improved CycleGAN structure and transfer learning with constraints, in order to learn the information from both domains to achieve better single image deraining results. However, these methods did not achieve the expected results. Based on this, a recent work [46] added bilateral constraint learning and contrast learning to adversarial learning, and achieved better deraining results under an unpaired rainy images dataset.

## 3. Proposed Approach

Though previous approaches improved performance, there are still many issues to be solved. These problems include that it is difficult to protect the structural information of the image, especially in complex situations. Previous approaches cannot reconstruct high-quality rain-free images well. Meanwhile, the network structures of previous approaches are not interpretable. Therefore, a method with strong interpretability, with the ability to effectively remove rain streaks and protect the structure of the image is in demand.

This section covers some basic concept relevant to the proposed approach, then the proposed model is introduced in detail.

### 3.1. Relevant Concepts

Discrete wavelet transformation (DWT) is a discretization of the scale and translation of the fundamental wavelet. 2D-DWT is usually implemented by four convolution filters with fixed parameters (convolution size of 2). After the DWT transformation, the image can be obtained as the four rightmost sub-bands shown in Figure 2. The output includes frequency domain information of the original image, while because of the downsampling operation, each sub-image also includes the spatial domain information of the original image. DWT is commonly used in image processing tasks, especially image compression, because in most cases the low-frequency components often already contain the basic features of the image, while the high-frequency signal only covers some detailed information of the image.

the DWT operation is suitable for the single image deraining task [47]. First, as rain streaks tend to be concentrated in high-frequency signals, and sub-bands containing information are contained in the frequency domain of the image after DWT, which will help the network to perform frequency selection and background information extraction in rain-free regions. Second, the spatial resolution of each band is reduced to half of original image after the DWT process. Therefore, the image is equivalently processed on a small scale, and the larger streaks in the original image become smaller, so the rain streaks are more easily removed. Third, the rain-free background area is often composed of large continuous pixels, a module with a larger receptive field is appropriate for extracting information, and DWT has a good balance between increasing the receptive field and the computing efficiency. In general, discrete wavelet transformation can capture both frequency domain and spatial location information, and because of its time-frequency localization property, it should be used to retain more detailed information, which will help to reconstruct the deraining images.

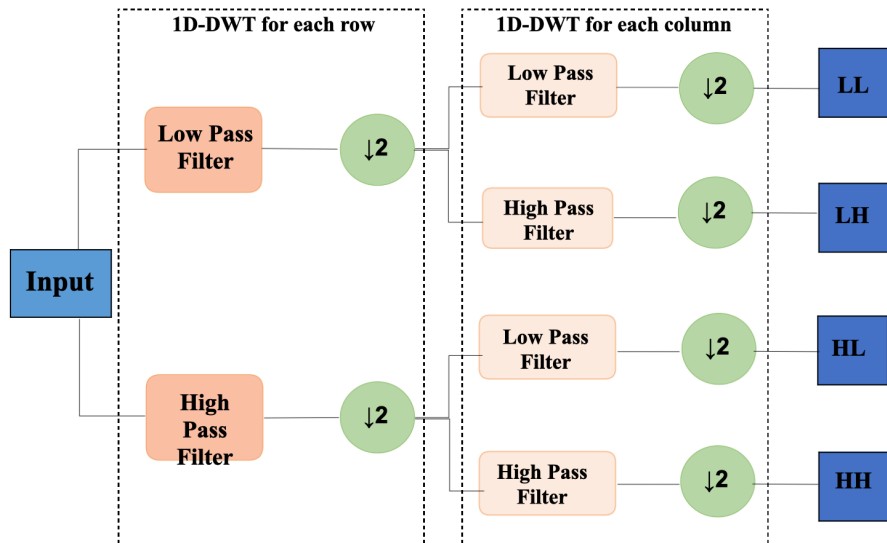

**Figure 2.** Diagram of the 2D-DWT process.

An attention mechanism is a structure that is widely embedded in network models [48,49]. It can autonomously learn a set of weight coefficients to represent the weight of importance. Currently, the attention mechanism has become one of the most widely used components in the field of deep learning. Similarly, attention mechanisms are also widely used in the field of computer vision, and the three main types of applications are channel attention, spatial attention, and self-attention. Channel attention is the most used in rain removal tasks, which can selectively use each channel's information and suppress useless features to reduce noise. Due to the powerful adaptive character of the attention mechanism, multiple channel attention modules are introduced to the proposed approach of this paper to adaptively guide the network in training. In addition to the commonly used channel attention module (SE-block), the Frequency Channel Attention [50] and Self-selective Kernel Attention [51] were also introduced to this proposed approach. They were utilized to decouple the high frequency background detail signals fused in the rain-obscured region and fused two-branch network structure.

Unlike the traditional channel attention module, the frequency channel attention (FCA) module promotes the global average pooling (GAP) in the frequency domain. To compensate for the shortcoming that GAP can only represent the lowest frequency information, FCA introduces a finite number of frequency components to replace the lowest frequency information. By integrating more frequency components, different information is extracted to form a multispectral description. Specifically, FCA can be derived based on sound mathematical theory, using a more general form of a two-dimensional discrete cosine transform (2D-DCT) instead of GAP to achieve fusion of multiple frequency components. This operation converts the spatial domain to the frequency domain, effectively increasing the frequency feature information and forming a multi-frequency channel attention, which can compensate for the shortcomings of insufficient feature information prevalent in traditional channel attention methods.

Selective Kernel Block, the twin of Squeeze-and-Excitation Block, is similar in structure to SE-block, the differences is that a selective kernel block is a multi-branch structure that can consider multiple receptive fields and thus has multiple scales. It was born from the joint inspiration of Inception-block and SE-block. It uses convolutional kernels with multiple different receptive fields to learn and obtain feature maps at different scales, first fusing the feature maps and then obtaining the weight coefficients of each branch feature separately through network learning, then performing weighted fusion on this basis and producing the final feature map. Thus, it takes into account the multiscale feature representation and allows the network to adaptively focus on the important scale features. Therefore, it can adaptively adjust the receptive fields and then dynamically reorganize the

features. The proposed approach is precisely the feature extraction through two branches with different receptive fields. Therefore, in order to refine the detailed signal in the rain-obscured area through the background signal in the rain-free area, and to perfectly fuse the two parts to obtain the final results, we introduced SK-block to help us better adaptively adjust the multi-branch fusion process.

### 3.2. Proposed Model

Single image deraining tasks often face problem of over smoothing for outcome images. This is due to the image detail appearing at a relatively high-frequency, and it is always coupled with high-frequency rain patterns [52–57]. However, approaches focusing on repairing the lost image background detail are rarely implemented. In order to extract the background from rain-free regions and recover details from rain-obscured regions, we proposed a new single image deraining network architecture based on frequency-aware rain scenes, which can be called FADNet (Frequency-Aware Deraining Network). The entire network structure with two branches is shown in Figure 3. The upper branch is used to extract the background details in the rain-free region with the low-frequency component, and the lower branch is used to repair the high-frequency details coupled in the rain-obscured region.

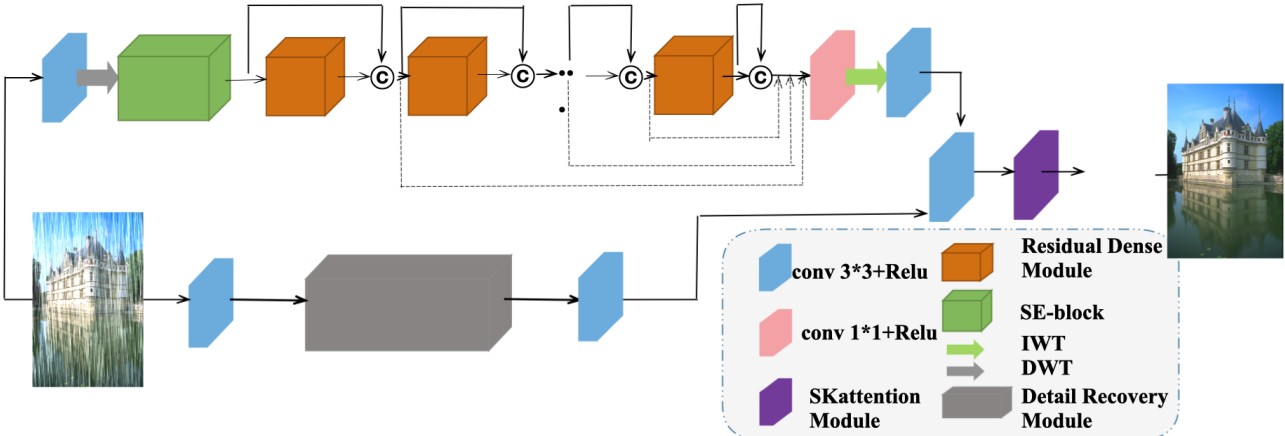

**Figure 3.** Diagram of the proposed approach.

As can be seen, discrete wavelet transform and channel attention block are introduced to extract the low-frequency background signal in the rain-free region more efficiently. In addition, the detail restoration module with frequency channel attention (FCA) is utilized to decouple the background detail signal information contained in the rain-obscured area adaptively. Then, SK-Block adaptively fuses the extracted two-branch signals, followed by a series of convolution operations to reconstruct a clear rain-free image.

The network acquires the information flow from different scales of the dual-branch. In addition to the introduction of DWT and SE-block in the upper branch, background signal extraction in the rain-free region is mainly performed by stacking the Residual Dense Module (RDM). After the DWT operation, the feature map of the low-frequency part of the frequencies is first passed through the SE-block and then input to the Residual Dense Network, which adaptively extracts the low-frequency background signal of the rain-free region. The residual dense network is composed of several residual dense blocks. This paper used three residual dense blocks considering the balance of performance and efficiency. Finally, the extracted residual dense network was connected to the other three partial frequencies in the original order and input to IWT losslessly to obtain the final background signal extraction network, which can be expressed as follows:

$$B = f_{iwt}([X\_LL', X\_LH, X\_HL, X\_HH]) \tag{1}$$

where *B* denotes the corresponding final clean image, $f_{iwt}$ denotes the inverse discrete wavelet transform operation, and $X\_LL'$ denotes the low frequency signal extracted by residual dense network operation.

The proposed model not only focuses on the background structure signal in the rain-free region, but also on the high-frequency background detail signal coupled in the rain-obscured region. In order to better preserve the background detail information in the rain-obscured region while removing the rain lines, the Detail Recovery Module is introduced in the following branches to decouple the background detail signals. The structure of this module is schematically shown in Figure 4.

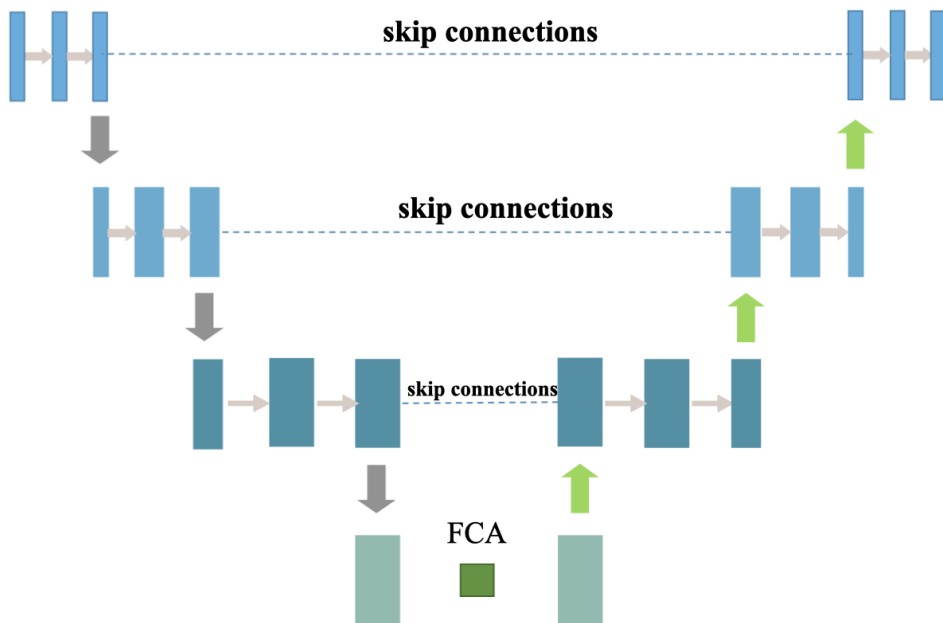

**Figure 4.** Diagram of the detail recovery module.

This figure shows that the detail recovery module adopts a network architecture similar to U-Net, with the frequency channel attention module embedded at the bottom. The process of this module is to first map the rain map to the feature potential space through the encoder of the U-Net, then perform different frequency band adaptive enhancement and weakening of the feature signals in the potential space through FCA, followed by extraction of the general signal of the rain-obscured region through the U-Net decoder which filters out the rain lines in the high frequency region. Finally, the detail signal from the rain-obscured region extracted in the high frequencies is further refined by performing the convolution operation passed into the fusion network combined with the background signal of the rain-free region. We can see that the detail recovery module adds a jump cascade between the coder–decoder, which allows the network to fuse the feature maps obtained from the encoders at the corresponding locations on the channel during the upsampling process. With this fusion function, the network is able to retain more detailed and generate clearer deraining images. The whole module contains three upsampling and downsampling instances, and each layer of the coder-decoder contains two convolution operations.

Unlike some previous algorithms that included two independent branch structures [53], this paper further considered the problem of information fusion with two branches. Considering that these two branches use different perceptual fields in the feature extraction process, they are independently distributed in the signal feature space with different sizes. To better combine the two signals, we introduced the SK-Attention module, which can dynamically select the receptive field size for a better fusion process. Specifically, feature signals obtained from the two branches are concatenated. Then, the features are input to

SK-block for multi-branch dynamic selection. The final deraining result is obtained after a series of convolution operations. The specific calculation process is as follows:

$$B = f_{conv}(f_{sk}(concat[f_1, f_2])) \tag{2}$$

where $f_1$ and $f_2$ denote the feature signals extracted from the two branches. $f_{sk}$ and $f_{conv}$ denote the SK-Block operation and a series of convolution operations, respectively. $B$ denotes the final deraining background image obtained after network reconstruction.

In this process, the upper branch generates the background signal from the rain-free region obtained. This background signal is utilized to refine the coupled signal of the rain-obscured region obtained from the lower branch. After that, the two parts are finally fused together. Because of the dynamic selection ability of SK-Block, multiscale image features are used efficiently to obtain better rain removal results.

## 4. Experiment Results and Analysis

This section covers the datasets and loss functions utilized in the experiment, experimental setup details, experimental results and analysis.

### 4.1. Experiment Datasets

The experiment was conducted on several commonly used deraining datasets. These datasets simulate different situations including light rain, moderate rain and heavy rain. These datasets include (1) Rain200L [35] which simulates the light raining situation, in which 1800 training pairs and 200 testing pairs are included; (2) Rain200H [35], which simulates the heavy raining and rain-mist situations, in which 1800 training pairs and 200 testing pairs are included; (3) Rain800 [58] which contains 700 training pairs and 100 testing pairs; (4) Rain1200 [59], which has 12,000 training pairs and 1200 testing pairs. For the 12,000 training pairs, 4000 pairs simulate light rain, 4000 pairs simulate moderate rain, and 4000 pairs simulate heavy rain. Table 1 shows the differences between these datasets. The sample rain streaks from these deraining datasets are shown in Figure 5.

**Table 1.** Comparison on commonly used deraining datasets.

| Datasets | Training Pairs | Testing Pairs |
|----------|----------------|---------------|
| Rain200L | 1800 | 200 |
| Rain200H | 1800 | 200 |
| Rain800 | 700 | 100 |
| Rain1200 | 12,000 | 1200 |



(a) Rain Streaks in Rain200L          (b) Rain Streaks in Rain200H          (c) Rain Streaks in Rain800

**Figure 5.** Sample rain streaks from deraining datasets.

### 4.2. Loss Function and Experiment Setup

Most of the deep learning-based deraining tasks utilize the mean squared error (MSE) as a loss function. MSE allots more weight to areas with a large difference, while areas with rain only have a small difference from high frequency details and do not have sufficient weight. Utilizing only MSE will lead to overlooking useful high-frequency information for

a deraining task. To obtain better results, the structural similarity index (SSIM) is added to the loss function, and the proposed loss function will be as follows:

$$L = |f(O) - B|^2 + 1 - \text{SSIM}(f(O), B) \tag{3}$$

In this equation, *B* indicates the ground truth image, and *f(O)* indicates the resultant image after deraining.

The experiment model is based on Pytorch. All the convolution layers have a 3*3 kernel, no batch normal layer is utilized, and the active function is ReLU. To be consistent for input and output images, the stride is set to 1, with 1 pixel padding. For the residual dense block, 3 blocks are utilized to balance the performance and efficiency. The model is trained by Adam, with a batch size of 16 and 300 epochs.

### 4.3. Results on Deraining Quality

Peak signal-to-noise ratio (PSNR) and structural similarity index (SSIM) are two of the most commonly used metrics to evaluate deraining images. PSNR is used to evaluate the color and brightness distortion, higher values indicate that a image looks more likely to reference image. SSIM is used to measure the structural similarity of tow images, two same image will have SSIM value of 1.

Ablation experiments were conducted on the Rain200H dataset, between Model 1, Model 2 and the proposed FADNet model. Model 1 has FCA but not DWT, Model 2 has DWT but not FCA, while the FADNet model has both DWT and FCA. The results of the ablation experiment are shown in Table 2. FADNet outperforms both Model 1 and Model 2. Specifically in PSNR, FADNet increases by 0.39 dB compared to Model 1 without DWT, indicating that DWT is an effective method to reconstruct clearer rain-free images. Additionally, for SSIM, FADNet increases by 0.07 compared to Model 2 without FCA, indicating that the FCA module can enrich the image details and help output images to maintain a clearer structure.

**Table 2.** Results of the ablation experiment for DWT and FCA modules.

| Approaches | With DWT | With FCA | PSNR-Rain200H | SSIM-Rain200H |
|---|---|---|---|---|
| Modle1 | NO | YES | 26.59 | 0.858 |
| Modle2 | YES | NO | 26.76 | 0.855 |
| FADNet (Proposed) | YES | YES | **26.98** | **0.862** |

Ablation experiments were also conducted on the Rain200H dataset to obtain the optimal number of residual dense blocks (RDBs). Results of the ablation experiment for the number of RDBs are shown in Table 3. The model has better performance when the number of RDB increases from 1 to 3. The models have similar performance with the RDB of 3, 4, and 5. When the number of RDB equals 3, the network achieved the best balance of performance and parameters. Therefore, FADNet selected three as the number of RDB.

**Table 3.** Results of the ablation experiment for numbers of RDB.

| RDB Numbers | PSNR-Rain200H | SSIM-Rain200H | Model Parameters |
|---|---|---|---|
| 1 | 24.88 | 0.846 | 1,473,955 |
| 2 | 26.56 | 0.858 | 1,768,995 |
| 3 (FADNet) | 26.98 | 0.862 | 2,064,035 |
| 4 | 26.99 | 0.860 | 2,359,075 |
| 5 | 26.95 | 0.861 | 2,654,115 |

The proposed approach (FADNet) was compared with several state-of-the-art approaches. These approaches include DSC [19], GMM [21], DDN [36], RESCAN [60], PReNet [61], SPANet [62], Syn2Real [63], MPRNet [52] and DID-GAN [46]. In these approaches, DSC and GMM are traditional deraining modes, the rest are popular deep learning-based deraining models.

Table 4 shows the deraining quality of various approaches. The proposed approach outperformed other approaches on most of the deraining datasets. Especially for the Rain800 dataset, the proposed approach achieved a large performance increase for both PSNR and SSIM. To make the comparison clearer, Figure 6 shows the PSNR value of different approaches, and Figure 7 shows the SSIM value of different approaches, while the last column indicates the proposed approach.

In addition to the PSNR and SSIM values, Figure 8 shows images after the deraining process. For Deraining images with the Rain200H dataset, the proposed approach produced less rain drops. The proposed approach also retained more details of the original image and possessed less blur.

**Table 4.** Comparison of the quality of deraining in deraining data sets.

| Approaches | PSNR-Rain800 | SSIM-Rain800 | PSNR-Rain1200 | SSIM-Rain1200 | PSNR-Rain200H | SSIM-Rain200H | PSNR-Rain200L | SSIM-Rain200L |
|---|---|---|---|---|---|---|---|---|
| DSC(2015) | 20.95 | 0.753 | 21.44 | 0.790 | 13.17 | 0.427 | 25.68 | 0.875 |
| GMM(2016) | 24.04 | 0.802 | 23.66 | 0.832 | 13.04 | 0.467 | 27.16 | 0.898 |
| DDN(2017) | 24.24 | 0.808 | 27.33 | 0.898 | 24.64 | 0.806 | 33.01 | 0.967 |
| RESCAN(2018) | 24.09 | 0.841 | 29.10 | 0.884 | 24.40 | 0.779 | 34.09 | 0.970 |
| PReNet(2019) | 24.90 | 0.806 | 30.17 | 0.900 | 25.52 | 0.854 | 32.41 | 0.914 |
| SPANet(2019) | 22.41 | 0.838 | 30.05 | 0.934 | 23.04 | 0.852 | 31.59 | 0.965 |
| MPRNet(2021) | 26.62 | 0.865 | 33.66 | 0.931 | **27.88** | **0.874** | 35.12 | 0.959 |
| FADNet(Proposed) | **27.49** | **0.886** | **33.78** | **0.941** | 26.98 | 0.862 | **35.68** | **0.972** |

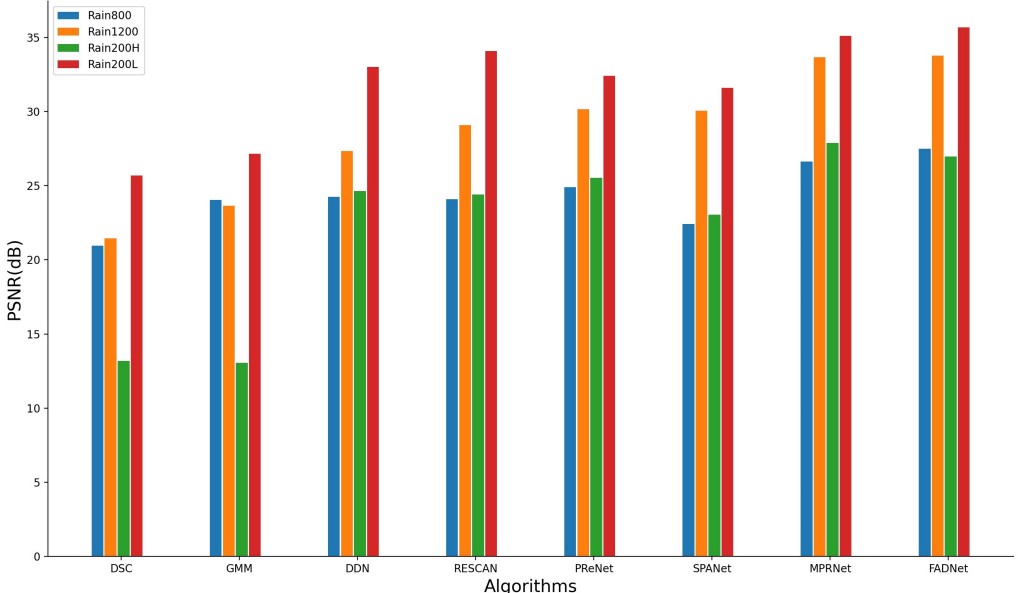

**Figure 6.** PSNR value of different approaches.

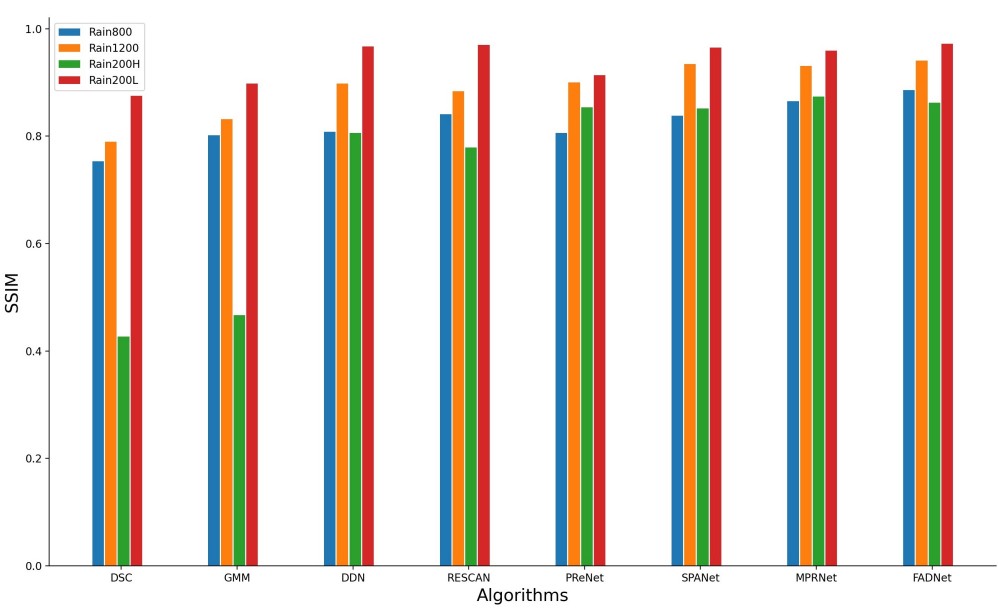

**Figure 7.** SSIM value of different approaches.

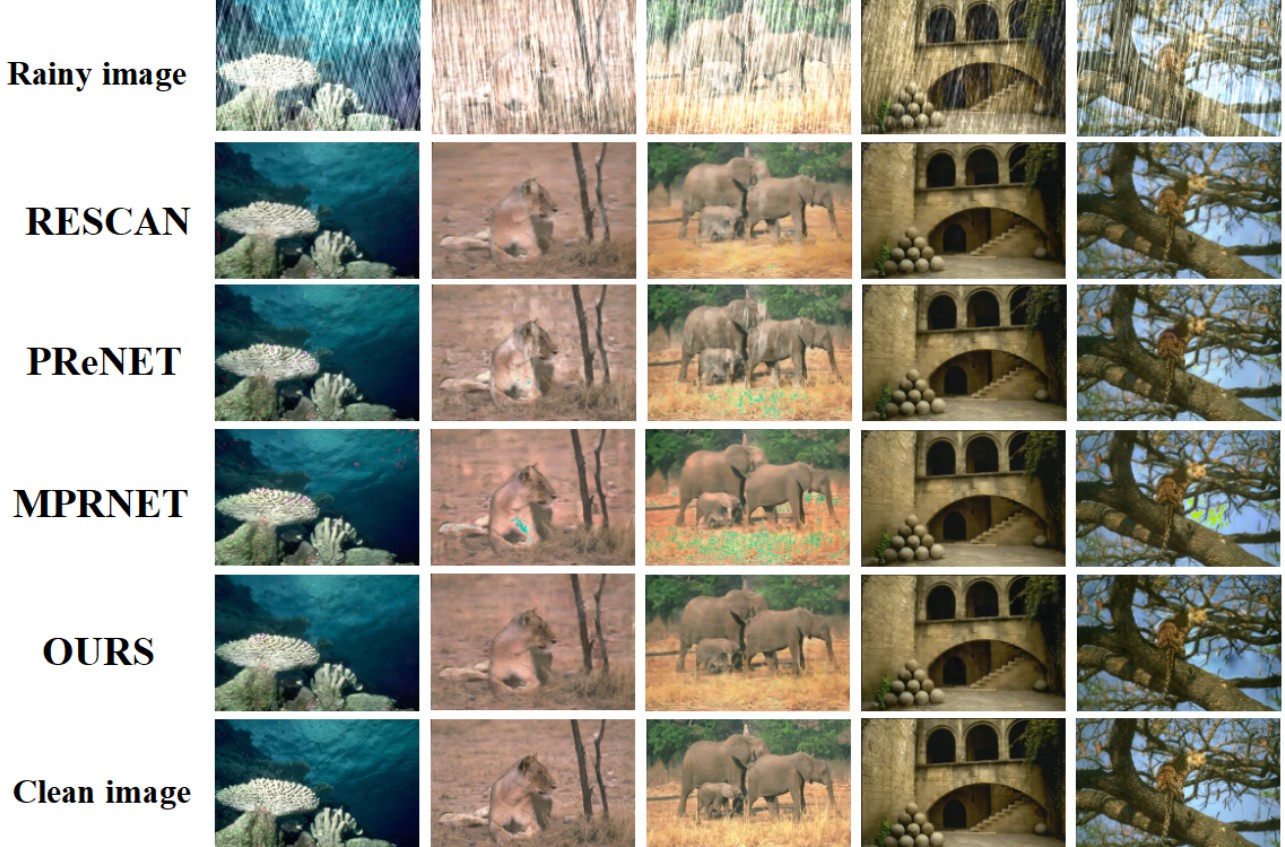

**Figure 8.** Results of rain images in Rain200H dataset.

### 4.4. Results on Time Consumption

Table 5 present the results for model parameters and processing time per image. For the model size, the proposed model had more parameters than the traditional model, but less parameters than the models proposed in recent years. For time consumption, the proposed approach was faster than most popular approaches.

**Table 5.** Comparison on model parameters and processing time per image

| Approaches | Model Parameters | Time (s) |
|---|---|---|
| RESCAN(ECCV'18) | 149,823 | 0.61 |
| PReNet(CVPR'19) | 168,963 | 0.19 |
| MPRNet(CVPR'21) | 3,637,249 | 0.26 |
| SPDNet(ICCV'21) | 3,318,741 | 0.28 |
| FADNet(Proposed) | 2,064,035 | 0.20 |

## 5. Conclusions and Discussion

This paper built a frequency-aware single image deraining model. The model had the following merits: (1) This model introduced DWT to extend the receptive field without extra computation cost. A clear low-frequency signal can be extracted by DWT. (2) Frequency channel attention is introduced to adaptively recover high frequency background details. (3) SK-Block is introduced to perform fusion from multi-receptive fields. The results on several deraining datasets indicate the efficiency and deraining quality of the proposed approach. The proposed approach has less artefacts and retains more useful details in an efficient manner.

Although the proposed FADNet achieved good results in effectively removing rain streaks, there is still room to improve. Future works will focus on improving the efficiency so that it can be used for real time application, and improving the generalization ability for real-world scenarios.

**Author Contributions:** Conceptualization, Y.H. and T.Z.; methodology, Y.H.; software, Y.X.; validation, Y.H., Y.X. and T.Z.; formal analysis, Y.X.; investigation, J.L.; resources, J.L. and H.W.; data curation, Y.H.; writing—original draft preparation, Y.H.; writing—review and editing, Y.H., J.L. and H.W.; visualization, Y.H. and J.L.; supervision, Y.H. and J.L.; project administration, Y.H. and H.W.; funding acquisition, H.W. All authors have read and agreed to the published version of the manuscript.

**Funding:** This research received no external funding.

**Institutional Review Board Statement:** Not applicable.

**Informed Consent Statement:** Not applicable.

**Data Availability Statement:** Not applicable.

**Conflicts of Interest:** The authors declare no conflict of interest.

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
