# Peer review of "Deep Leaning Based Frequency-Aware Single Image Deraining by Extracting Knowledge from Rain and Background"

_make, doi:10.3390/make4030035_

Round 1
Reviewer 1 Report
This paper presents a method for single image deraining using a deep neural network. The idea is to consider the high and low frequency components in the image separate the foreground rain region and background scene regions. The network structure design is based on discrete wavelet transform. Some experimental results are reported by testing on the public datasets. The performance evaluation is carried out by comparing with several existing algorithms. Although the method is described and results are provided. There are several issues to be addressed. First, although the DWT is adopted, but it is not clear how its role in the network structure. In fact, as shown in Figure 3, DWT is only used for providing data from one layer to another. The contribution seems not clear and needs to have more detailed description. Second, to separate the foreground and background information, it is necessary to consider the spatial variant case, such as the works "Lin, H.Y., Li, K.J. and Chang, C.H., 2008. Vehicle speed detection from a single motion blurred image. Image and Vision Computing, 26(10), pp.1327-1337" and "Wu, Q., Chen, L., Ngan, K.N., Li, H., Meng, F. and Xu, L., 2020, December. A Unified Single Image De-raining Model via Region Adaptive Coupled Network. In 2020 IEEE International Conference on Visual Communications and Image Processing (VCIP) (pp. 1-4). IEEE". The mixture of the rain and background scene needs to be further discussed. Third, the network structure is fairly simple, it is not clear what are the important parts with most contributions on deraining. The authors should provide the ablation study for verification. Fourth, the article is very difficult to read. There are lots of grammar mistakes through the paper. It is required to have substantial rewriting.
Reviewer 2 Report
The research paper proposes a novel frequency-aware single image deraining network via separation of rain and background for efficiently obtaining rain removal images with more detailed information. For the rainy images, most of the background key information belongs to the low frequency components and the high frequency components are mixed by background image details and rain. In contrast with other related papers, the proposed network has 3 major contributions: 1) a proposed residual dense network based on DiscreteWavelet Transform (DWT) for studying the rainy image background information, 2) an introduced frequency channel attention module into adaptive decoupling of high-frequency image detail signals, as well as 3) a fusion module containing the attention mechanism to make full use of the two-branch of the multi-scale information, using the context information in large area.
The proposed approach is evaluated by several representative datasets and experimental results depict that this proposed approach outperforms other state-of-the-art deraining algorithms.
This seems to be a very challenging topic and authors have solidly proven their findings. Nevertheless, some points need to be clarified before this submission can be accepted.
More specifically, the preliminaries, the contribution, the definitions, the algorithms as well as the mathematical types are detailed in a thorough way showing readers the potential of this area; also, the research appears to be complete in terms of thorough explanations.
On the other hand, I think that the paper should ideally be enriched with some figures in terms of results so as to further explain to users who are not very familiar with this field, the purpose of appreciating the precise contribution made by this paper (by looking at tables it is more difficult to understand this contribution).
Authors have also omitted to discuss the problem at hand. Also, they have not stated the differences of their work with others in bibliography.
Moreover, could authors add additional features or even other algorithms in order to better fine-tune their experiments?
The datasets analysis is somehow missing. Please provide additional datasets characteristics in terms of tables.
Finally, authors should add a paragraph regarding the Future Work.
Round 2
Reviewer 1 Report
This revision has addressed most of the reviewer's concerns, including the experiments, ablation study, etc. It is now in an acceptable form, but still need some work to polish the English writing. In addition, it is highly suggested to provide the code associated with this paper to increase the impact of the paper and journal.
This manuscript is a resubmission of an earlier submission. The following is a list of the peer review reports and author responses from that submission.